# Complete Remission of Mouse Melanoma after Temporally Fractionated Microbeam Radiotherapy

**DOI:** 10.3390/cancers12092656

**Published:** 2020-09-17

**Authors:** Cristian Fernandez-Palomo, Verdiana Trappetti, Marine Potez, Paolo Pellicioli, Michael Krisch, Jean Laissue, Valentin Djonov

**Affiliations:** 1Institute of Anatomy, University of Bern, 3012 Bern, Switzerland; cristian.fernandez@ana.unibe.ch (C.F.-P.); verdiana.trappetti@ana.unibe.ch (V.T.); marine.potez@gmail.com (M.P.); jean-albert.laissue@pathology.unibe.ch (J.L.); 2Biomedical Beamline ID17, European Synchrotron Radiation Facility, 38043 Grenoble, France; paolo.pellicioli@esrf.fr (P.P.); krisch@esrf.fr (M.K.)

**Keywords:** synchrotron microbeam radiation therapy, temporal fractionation, mouse, melanoma, spatial fractionated radiotherapy, melanophages, macrophages

## Abstract

**Simple Summary:**

Synchrotron Microbeam Radiation Therapy (MRT) provides excellent tumour control in preclinical cancer models. Most of the results obtained in the field employ average peak doses between 300 and 600 Gy. However, despite its proven efficacy, one of the biggest challenges in translating MRT to the clinic is that these high doses can be delivered only at synchrotron facilities, and there are only a few in the world. To solve this problem, we delivered three sessions of lower peak doses of 133 Gy, which approach clinical capabilities. The results showed complete tumour remission for 18 months. In fact, it ablated 50% of the primary melanomas with no metastasis. We believe that this study is of value because (i) it increased the therapeutic index of MRT, and (ii) it demonstrates that MRT can be less dependent on synchrotron sources, which opens the path for future clinical application.

**Abstract:**

Background: Synchrotron Microbeam Radiotherapy (MRT) significantly improves local tumour control with minimal normal tissue toxicity. MRT delivers orthovoltage X-rays at an ultra-high “FLASH” dose rate in spatially fractionated beams, typically only few tens of micrometres wide. One of the biggest challenges in translating MRT to the clinic is its use of high peak doses, of around 300–600 Gy, which can currently only be delivered by synchrotron facilities. Therefore, in an effort to improve the translation of MRT to the clinic, this work studied whether the temporal fractionation of traditional MRT into several sessions with lower, more clinically feasible, peak doses could still maintain local tumour control. Methods: Two groups of twelve C57Bl/6J female mice harbouring B16-F10 melanomas in their ears were treated with microbeams of 50 µm in width spaced by 200 µm from their centres. The treatment modality was either (i) a single MRT session of 401.23 Gy peak dose (7.40 Gy valley dose, i.e., dose between beams), or (ii) three MRT sessions of 133.41 Gy peak dose (2.46 Gy valley dose) delivered over 3 days in different anatomical planes, which intersected at 45 degrees. The mean dose rate was 12,750 Gy/s, with exposure times between 34.2 and 11.4 ms, respectively. Results: Temporally fractionated MRT ablated 50% of B16-F10 mouse melanomas, preventing organ metastases and local tumour recurrence for 18 months. In the rest of the animals, the median survival increased by 2.5-fold in comparison to the single MRT session and by 4.1-fold with respect to untreated mice. Conclusions: Temporally fractionating MRT with lower peak doses not only maintained tumour control, but also increased the efficacy of this technique. These results demonstrate that the solution to making MRT more clinically feasible is to irradiate with several fractions of intersecting arrays with lower peak doses. This provides alternatives to synchrotron sources where future microbeam radiotherapy could be delivered with less intense radiation sources.

## 1. Introduction

Synchrotron Microbeam Radiation Therapy (MRT) is a preclinical approach of spatial fractionation that has been found to obtain superior tumour control in different animal models relative to a homogenous radiation field [1,2,3]. Synchrotron-based MRT uses a multi-slit collimator to spatially fractionate synchrotron X-rays into an array of microbeams that range from 20 to 100 μm in width, which are spaced by 50–500 μm from beam centre to beam centre [4]. This spatial arrangement of the radiation beam allows a sharp transition between high peak-dose deposition in the microbeam path (typically 300–800 Gy) and the region between these microbeams (valley), which typically receives 1–10% of the peak dose [4]. These doses are delivered in a single fraction at ultra-high dose rates (up to 16,000 Gy/s in the European Synchrotron) [4]. As a result, MRT induces extremely low normal-tissue radiation toxicity, which is attributed, in part, to the rapid delivery of radiation (<200 ms), and this is known as the “FLASH” effect [5]. Moreover, normal tissue tolerates the high peak doses in accordance with the dose–volume effect principle, which is exploited by the spatially fractionated geometry of MRT [6,7,8,9,10].

Regarding MRT’s anti-tumour effects, the published literature shows that MRT targets the tumour microenvironment with exceptional efficacy and unique complexity: MRT increases tumour cell death when compared to a homogenous beam of synchrotron radiation [11], selectively disrupts immature tumour blood vessels [12], produces transient vascular permeability [13,14,15], induces synchrotron-specific bystander effects [16,17,18,19], and triggers an influx of inflammatory cells into the irradiated tumour [15]. However, even if future MRT clinical trials at synchrotron-based facilities were successful, the dependence on synchrotrons to deliver high peak doses would hamper its deployment in hospitals. To bridge this gap, it was decided to explore the use of lower peak doses by delivering three fractions of 133.41 Gy and compare its anti-tumour efficacy against a single MRT of 401.23 Gy.

Exploring the benefits of temporally fractionating MRT was one of the aims of this manuscript. Temporal fractionation in conventional radiotherapy is known to provide better tumour control while producing lower normal tissue toxicity than a single large dose; it increases tumour damage by allowing both the redistribution of tumour cells into more vulnerable phases of the cell cycle (late G2, and M phase) [20] and the reoxygenation of tumour cells, thereby increasing their radiovulnerability [21]. At the same time, normal tissue toxicity is lower because there is time for the repair of sublethal damage and also for the repopulation of cells between fractions [22]. We hypothesised that MRT would benefit from temporally fractionating the dose and thus also allowing the delivery of lower peak doses.

This manuscript demonstrates that three sessions of temporally fractionated MRT intersected at 45 degrees over 3 days were able to ablate 50% of the B16-F10 mouse melanomas. Moreover, the cured mice did not develop metastasis and had no behavioural abnormalities for 18 months. The histopathological assessment of the cured mice revealed that in the place where the tumour was located, a high concentration of melanin-laden macrophages was found, indicating that the immune system played a role in tumour eradication.

## 2. Results

### 2.1. Tumour Growth

Two groups of 12 mice harbouring B16-F10 melanomas in their ears were exposed to either a single MRT fraction (Figure 1A) or to three intersecting MRT arrays delivered once per day over three consecutive days (Figure 2A). The single MRT array delivered a peak dose of 401.23 Gy with a valley dose of 7.4 Gy, while the fractionated MRT delivered peak doses of 133.41 Gy with valley doses of 2.46 Gy (with an accumulative dose equal to the single MRT). The second and third fractions were delivered at 45° with respect to the preceding one (Figure 2A).

Figure 1B shows the growth of all B16-F10 melanomas exposed to single MRT (1 × 401.23 Gy). The growth of these tumours was retrospectively analysed according to their response and further classified into two post-treatment categories (Figure 1C): (i) 66.6% exhibited tumour shrinkage and consecutive regrowth with an average growth delay of 9.8 days (responsive tumours, in blue), while (ii) 33.3% did not display tumour shrinkage (non-responsive tumours, in red).

In contrast, Figure 2B shows the growth of all B16-F10 melanomas exposed to three fractions of MRT (3 × 133.41 Gy). These tumours were also retrospectively analysed and re-classified into three distinct categories (Figure 2C): (i) those that experienced complete remission (50%; ablated tumours, in green), (ii) those that exhibited a mean growth delay of 12.8 days (37.5%; responsive tumours, in blue), and (iii) those that did not show tumour shrinkage (12.5%; non-responsive tumours, in red).

### 2.2. Animal Survival

Mice exhibiting complete remission did not show behavioural abnormalities, and no organ metastasis were detected. We ended the experiment 18 months after treatment (540 days) and during this time, the mice did not have behavioural abnormalities nor signs of cancer. Figure 3A depicts animal survival with results indicating that the treatment response favours the triple fractionation of MRT, with statistical significance computed by the Gehan–Breslow–Wilcoxon test, which gives extra weight to early timepoints (*p* < 0.001). Further analysis revealed a 2.5-fold increase in the median survival after three fractions of MRT (Figure 3B). Figure 3 also includes data from our previously published experiment [3] showing the survival of mice harbouring unirradiated melanomas (<unirradiated>). Taking into consideration the growth of unirradiated tumours, the temporally fractionated MRT scheme increased the median survival by 4.1-fold.

### 2.3. Histopathology of Ablated Tumours

During the daily tumour follow-up, we observed that when the melanomas shrank, they lost all of their volume and became completely flat. Moreover, they did not regrow, and the ears displayed a distinct black pigmentation of melanin on the skin, which resembled a nevus (Figure 4A). At the early stages of the experiment, we hypothesised that the melanomas were senescent with the potential to re-grow at any time. However, this did not happen, and the mice aged without complications. Upon histopathological analysis, the results revealed that the melanin was contained inside the cytoplasm of large cells, which are distinct from the rest of the cells in the skin (Figure 4B).

The absence of melanoma cells in the ears was confirmed with immunohistochemistry. The melanin-laden cells were negative for two melanoma markers: the cytoplasmatic staining of Melan-A (Figure 4E) and the nuclear staining of SOX10 (Figure 4G). Instead, the melanin-laden cells were positive for the macrophage marker CD68 (Figure 4C), revealing that the “dormant tumour” was in fact a large group of melanophages. As a positive control, we used an untreated melanoma that was harvested 10 days after implantation. Positive tumour-associated macrophages appear in red in Figure 4D. Clearly identified melanoma cells with the cytoplasmatic staining of Melan-A appear in red (Figure 4F), and clearly identified melanoma cells with the nuclear staining of SOX10 also appear in red (Figure 4H).

Analysis of the 18-month-old nevus with electron microscopy revealed that melanophages were heavily loaded with melanin (Figure 4I), while tumour-associated macrophages from a melanoma harvested 6 days after MRT showed only a few granules (Figure 4J). Moreover, the nevus was formed by an agglomerate of melanophages (macrophages densely packed with phagocytosed melanin granules, which appear as dark dots in the cytoplasm). It can also be observed that the melanophages are separated by connective tissue containing dense collagen fibres, indicating that these cells had been there for an extended period of time. In contrast, tumour-associated macrophages seen 6 days after MRT (Figure 4J) show loose cytoplasm and swollen mitochondria (early stage of cell death), as well as numerous primary and secondary lysosomes. The melanoma cells look viable and contain a few melanin granules.

## 3. Discussion

The data in this manuscript demonstrate that three sessions of temporally fractionated MRT arrays intersecting at 45 degrees and delivered over 3 days had the unique capability of eradicating 50% of the primary melanomas and preventing organ metastases for 18 months. In the rest of the animals, this treatment scheme increased the median survival time 2.5-fold (relative to the single MRT fraction) and 4.1-fold against untreated tumours. The fractionated MRT scheme was also associated with a delay in tumour growth of 27% (from 9.8 to 12.8 days) and a reduction of 20.8% in the number of tumours that did not respond to the treatment (from 33.3 to 12.5%).

The benefits of temporally fractionating the dose are well-known in conventional radiotherapy but have been scarcely explored in MRT. In 2009, Serduc and colleagues exposed 9 L gliosarcoma-bearing rats to three MRT arrays composed of microbeams 50 μm wide and 211 μm from beam centre to beam centre, with a dose rate of approximately 16,000 Gy/s at the European Synchrotron [23]. Tumours received peak entry doses of 400 Gy on the first day, 360 Gy on the second day, and 400 Gy on the third day. The MRT arrays were multidirectional, intersecting at 90 degrees over three days, with the valley dose being approximately 15 Gy each day at the tumour centre. Under these configurations, the authors reported an increase of 216% in the lifespan of the rats against those with untreated tumours. However, some rats died from tumour recurrence. It is important to point out that the authors used a different tumour and animal model, and they intersected three multidirectional arrays, while we intersected three unidirectional arrays. These two differences could explain their less auspicious results with respect to ours. In addition, the authors did not use a single accumulated MRT dose for comparison. In another study published in 2011, Uyama and colleagues treated human glioma xenografts in mice over 2 days with cross-fired MRT [24]. The authors intersected two unidirectional MRT arrays at 90 degrees on Day 1 and repeated on Day 2. The MRT array was composed of microbeams 100 μm wide separated from their centres by 500 μm. The peak dose was 130 Gy per array, the cumulative valley dose was 9.6 Gy per day, and the dose rate was 124 Gy/s. Under these configurations, the authors observed a strong tumour growth suppression of 86.5% relative to the unirradiated control group, but the median survival of the animals was not reported. Again, the authors used a different tumour model and did not compare tumour growth suppression to a single accumulated MRT dose. In the present study, we demonstrate that our fractionated MRT scheme (3 × 133 Gy peak dose intersected in 45 degrees over 3 days) ablated the primary melanomas, whereas delivering the total cumulative dose in a single MRT fraction (1 × 401 Gy peak dose) did not. When comparing the valley doses, one can notice that the studies from Serduc et al. and Uyama et al. had valley doses of 15 Gy/day (×3 days) and 9.6 Gy/day (×2 days) respectively, which were higher than the valley dose that we used of 2.46 Gy/day (×3 days). The implications of valley doses are an important variable to keep in mind, since most valley doses would be considered significant in a radiobiology context on their own, and future work should focus on elucidating the role of the valley dose on every tumour model.

The geometry of the MRT array may also significantly influence tumour control. Specifically, Uyama and colleagues [24] spaced the microbeams every 500 μm, which was considerably larger with respect to the 211 μm used by Serduc et al. [23] and the 200 μm employed here. Interestingly, Uyama et al. used microbeams of 100 μm in width, while we (together with Serduc et al.) used microbeams of 50 μm. When comparing the ratio of microbeam width to the spacing between the microbeams of each array used in these studies, the values vary: 1:5 for Uyama et al., 1:4.2 for Serduc et al., and 1:4 for us. In light of the tumour remission obtained with our fractionated MRT scheme, this suggests that a short ‘microbeam-to-spacing ratio’ may significantly improve tumour control.

Temporal fractionation of the dose also influences the tumour response. We have shown that a single peak dose of 400 Gy of MRT markedly reduced the functional vascularity and blood perfusion of the B16-F10 melanomas [3]. The three peak doses of only 133 Gy used in the present study could be inducing similar vascular disruptions. This particular effect is likely to become especially relevant for the second and third MRT fractions because of repetitive reinforcement of the damage to the vascular network of the melanoma. In fact, studies from Song et al. performed in murine tumours but after semi-ablative high-dose hypofractionated radiotherapy (10–30 Gy) showed a marked increase in tumour hypoxia [25]. The authors also revealed that the proportion of surviving hypoxic cells decreased post-irradiation, which was most likely due to the reoxygenation triggered by the decrease in oxygen consumption caused by the massive death of tumour cells. Reoxygenation may have also played a crucial role during our experiments, and future studies should focus on quantifying the influence of hypoxia after MRT.

The delivery of the MRT array in different anatomical planes is also a factor that needs consideration. In fact, in the present study, the proportion of tissue exposed to peak and valley doses changed as the number of intersecting MRT arrays increased. Figure 5 shows the approximate geometrical distribution of the peak and valley doses from one to three arrays. By looking at the figure, it is evident that as the MRT arrays intersect, more tissue is targeted by the peak doses, while less tissue remains exposed to the valley doses. This phenomenon could be explored further by intersecting more than three arrays (i.e., from more than three ports). Based on the present results, we hypothesise that the antitumoural effects of intersecting more arrays will be even higher. However, the delivery of more arrays from only one direction (one port) will most definitely augment normal tissue toxicity and limit the number of possible intersections, whereas targeting the tumour with arrays converging from several directions (several ports) would largely prevent that. A limitation of this study was that we did not include a group of mice exposed to the three intersecting arrays on the same day, although this would have abolished the benefits of temporally fractionating the dose (i.e., reoxygenation, normal tissue repair, etc.). Nevertheless, we plan to explore this avenue when the logistical problems due to Covid-19 abate.

The histopathological assessment of the cured mice revealed a high concentration of melanin-laden macrophages where the melanoma was located. This suggests that the immune system may have played a role in the elimination of the tumour as published data show that fractionated radiotherapy (but not a single fraction) can sometimes trigger systemic anti-tumour immune responses, which is a phenomenon also mentioned in the literature as abscopal effects [26]. Preclinical studies on breast and colon carcinoma showed that three fractions of 8 Gy or five fractions of 6 Gy on consecutive days (in combination to CTLA-4 blockade) were superior at inducing abscopal effects than a single ablative dose of 20 Gy [26], demonstrating that time is critical when it comes to eliciting abscopal effects. Although we lack evidence that abscopal effects occurred in our experiments, it is plausible that a connection could exist with the melanophages observed in the ablated tumours. Future mechanistic studies should focus on investigating the involvement of the immune system after MRT and also on examining the histopathology of tumours at early timepoints. Overall, we attribute our results to (i) a higher volume of tumour tissue exposed to peak doses as a result of the cross-fired arrangement over the 3 days, (ii) tumour reoxygenation, and (iii) the possible occurrence of anti-tumour immune responses triggered by the fractionated scheme but not by the single fraction.

## 4. Materials and Methods

### 4.1. Animal and Tumour Model

Eight to ten-week-old young-adult female C57BL/6J mice (Charles River, L’Arbresle, Rhône, France) weighing 18–20 g on arrival were housed and cared for in the animal facility of the European Synchrotron at 50% humidity following a 12 h day/night cycle. Two weeks after their arrival, mice were subcutaneously implanted with 120,000 B16-F10 melanoma cells in both ears following our published protocol [27]. The in-ear melanomas are easily visible by the naked eye at the time of irradiation. Therefore, this tumour model does not require any image guiding system. The experiment was terminated when one of the following humane endpoints were reached: either the melanoma reached a size of 100 mm^3^ or the melanoma presented skin laceration due to its rapid growth. All the procedures presented here were approved by the Swiss Animal Welfare Office from the Canton of Bern under the permit number BE61/15 and by the French Ministry of Higher Education and Research under the permit number APAFIS#9605-2017041013465762 v2.

### 4.2. Culture of B16-F10 Cells

The B16-F10 melanoma cells originated from the C57BL/6J mouse strain and were bought from the American Type Culture Collection (ATCC-CRL-6475, LGC Standards GmbH, Wesel, Germany). They are adherent cells with a mixed morphology of epithelial-like and spindle-shaped cells. In preparation for the tumour implantation, the B16-F10 cells were thawed 2 weeks prior to the experiments, passaged every 3–4 days or when reaching 90% confluency, and maintained in growth medium composed of 0 UK Origin; supplemented with 10% certified Fetal Bovine Serum (FBS) (cat: 16000044, Gibco^®^, Grand Island, NY, USA). On the day of implantation, flasks containing B16-F10 cells on the logarithmic-growth phase were selected; they had been given a medium change 24 h before the experiment. Then, cells were washed with Dulbecco’s phosphate-buffered saline(DPBS), detached with 5 mL of Trypsin-Ethylenediamine Tetraacetic Acid (EDTA) (0.25%), and the trypsin was neutralised with 10 mL of fresh growth medium. The cell suspension was immediately centrifuged for 5 min at 130 rcf (relative centrifugal force) and the pellet was re-suspended in 1 mL of Dulbecco’s Modified Eagle Medium (DMEM) FBS-free. Cells were counted with a disposable Hemocytometer C-Chip (Model: DHC-N01, NanoEntek Inc., Seoul, Korea), centrifuged once more, and re-suspended with the correct amount of DMEM FBS-free medium that allows having a cell concentration of 120,000 cells in 4 μL. The final cell suspension was placed on ice and proceeded immediately with the tumour implantation.

### 4.3. Tumour Implantation

In preparation for the microsurgery, each mouse was anesthetised via an intraperitoneal (IP) injection with an agonist cocktail composed of 0.05 mg/kg of body weight (BW) of fentanyl, 5 mg/kg of BW of midazolam, and 0.5 mg/kg of BW of medetomidin; all were diluted in sodium chloride 0.9%. The implantation of the melanoma is made on the ventral side of the mouse ears. Three fine cuts are made on the ear’s skin along the superior, inferior, and lateral side of a square. Then, the skin is detached from the cartilage (from lateral to medial) through a movement that resembles opening a door. Then, cuts are made on the exposed cartilage following the four sides of a square. Then, the cut square-cartilage is removed, leaving a window to the dorsal side on the skin of the ear. A clot of 120,000 B16F10 cells is placed inside this window. The clot is formed with 3 µL of fibrinogen (5 mg/mL), 3 µL of thrombin (3 mg/mL), and 120,000 cells suspended in 4 µL of cell culture medium. The skin from the ventral side is immediately repositioned back (the door is closed), and the three-ventral skin cuts are sealed using Topical Skin Adhesive (Histoacryl^®^, B. Braun Medical AG, Sempach, Switzerland). After the microsurgery, each mouse was immediately labelled on its tail using coloured markers, and the anaesthesia was reversed via an IP injection of the following antagonist cocktail: 0.5 mg/kg of BW of flumazenil, 2.5 mg/kg of BW of atipamezol, and 0.1 mg/kg of BW of buprenorphine; all diluted in sodium chloride 0.9%. The measurement of the tumour size was performed daily starting on day 7 after tumour implantation. A digital calliper was used to measure the size of each tumour following the three axes (width, height, depth). Details of the protocol can be found in our previously published paper [27].

### 4.4. Irradiation and Dosimetry

The irradiations started 9 days after implantation, and the mean volume of the melanomas was 11.3 mm^3^ (SD = 5.4). Twenty-four mice were distributed among two experimental groups of 12 animals. The melanomas were exposed to either a single MRT array or three unidirectional MRT arrays that were intersected in 45 degrees over 3 days.

The delivered dose was calculated using Monte Carlo simulations based on Geant4 toolkit [28]. The mouse ear was modelled as a disk of water with a 0.3 mm thickness and 5 mm radius. The tumour was created in the centre of the disk as a sphere of water with the same mean volume of the measured melanomas. The 7.6 × 8.0 mm^2^ radiation field used was composed by an array of 38 vertical microbeams, 50 µm wide and spaced 200 µm from their centres. Considering a volume of 1 mm^3^ in the centre of the tumour, a peak-to-valley dose ratio (PVDR) of 54.24 was found (Table 1). The calculation of output factors relative to dosimetry under reference conditions [29] allowed the determination of the dose delivered in the melanoma. The single MRT array delivered a peak dose of 401.23 Gy, while the three sessions of MRT delivered peak doses of 133.41 Gy on each session (with an accumulative dose similar to the single MRT). The corresponding valley doses were 7.40 Gy and 2.46 Gy for the single session and three sessions, respectively (Table 1).

All irradiations took place at the ID17 Biomedical Beamline of the European Synchrotron Radiation Facility (ESRF) in Grenoble, France. As shown in Table 1, the polychromatic photon spectrum used during the irradiations had a mean energy of 104 keV with an intensity peak at 88 keV and a dose rate of 68.99 Gy/s/mA. With an available synchrotron storage ring current between 168 and 200 mA, irradiations were performed at the mean dose rate of 11,704 Gy/s for the single MRT and 12,870 Gy/s for the fractionated MRT. Before animal treatments, the irradiation geometry was verified using a radiochromic film placed on the mouse ear.

### 4.5. Histopathology and Immunohistochemistry

After tissue harvesting, the ears bearing melanomas were collected and fixed according to the following steps. Fixation with 4% paraformaldehyde in PBS over 24 h (the volume of fixative was at least 10× the volume the ear), three washes in PBS, removal of fixative with 15% glucose over 6 h, three washes in PBS, and 70% ethanol until paraffin embedding. Serial sections of 4 μm were prepared and stained with haematoxylin and eosin. Upon observation of the melanin-laden cells, we proceeded to perform immunohistochemistry for the following sets of primary antibodies: (i) Rabbit monoclonal anti-MelanA (ab210546, Abcam, Cambridge, UK) diluted with 3% milk at 1:1000, (ii) Rabbit monoclonal anti-CD68 (ab125212, Abcam, Cambridge, UK) diluted with 3% milk at 1:500, and (iii) Rabbit monoclonal anti-SOX10 IgG (cat: 12-1244, Abeomics, San Diego, CA, USA) diluted with 3% milk at 1:200. The second antibody was the ImmPRESS goat anti-rabbit Immunoglobulin G polymer (Vector Labs, MP-7451, Burlingame, CA, USA) diluted with ImmPRESS reagent at 1:27.

Briefly, our general protocol was the following: deparaffinisation of the slides, heat-induced antigen retrieval for 20 min (buffer: 1 mM EDTA with 0.05% Tween 20, pH8), cooling at room temperature for 20 min, wash in PBS, 0.5% Tween 20 in PBS for 30 min, and two washes in PBS. Blocking steps included: blocking with 2.5% normal goat blocking solution for 20 min at room temperature, blocking with 3% milk for 20 min at room temperature, and protein block (DAKO, Agilent Technologies, Santa Clara, CA, USA) for 20 min at room temperature. Immediately after the slides were incubated with the primary antibody for 1 h at 37 °C, they were rinsed with PBS for 5 min, incubated in 0.6% H_2_O_2_ in methanol for 30 min at room temperature, washed twice in PBS for 3 min, incubated with secondary antibody for 30 min at room temperature, washed three times in PBS for 5 min, incubated in a solution of 3,3′-Diaminobenzidine (DAB) for 8 min at room temperature, and rinsed in distilled water for 5 min. Then, we proceeded to bleach the melanin by incubating the sections in 1% trichloroisocyanuric acid (TCCA) (freshly made) for 60 min at room temperature. Slides were counter-stained with Myer’s hemalum (1:1) for 2–3 min, dehydrated, and covered. The slides from the control samples were not bleached, since the amount of melanin did not interfere with the visualisation of the antibodies. In addition, the slides were stained with the NovaRED substrate kit for peroxidase Reagent (MP-7451, Vector Labs, Burlingame, CA, USA) instead of DAB.

## 5. Conclusions

In conclusion, the present study demonstrates that our fractionated MRT scheme intersected at 45 degrees angle over 3 days ablated 50% of the primary melanomas, whereas delivery of the total cumulative dose in a single MRT fraction did not. The fractionated scheme also increased by 2.5-fold the survival of mice in comparison to the single MRT session, and by 4.1-fold against the untreated mice (reused data from our previously published study [3]). The exact radiobiological mechanisms involved in synchrotron MRT are not completely understood, but we attribute our results to the temporal fractionation of the MRT arrays, the cross-fired arrangement of the MRT arrays, and the possible occurrence of anti-tumour immune responses. Nevertheless, certain technical parameters still need to be optimised: peak dose per fraction, number of fractions and planes of delivery, and frequency of fractions. For now, what is clear is that high peak doses delivered in a single fraction are not a pre-requisite for successful MRT, suggesting that a more clinically feasible delivery method is possible and as a result makes this therapy less dependent on synchrotron facilities.

## Figures and Tables

**Figure 1 cancers-12-02656-f001:**
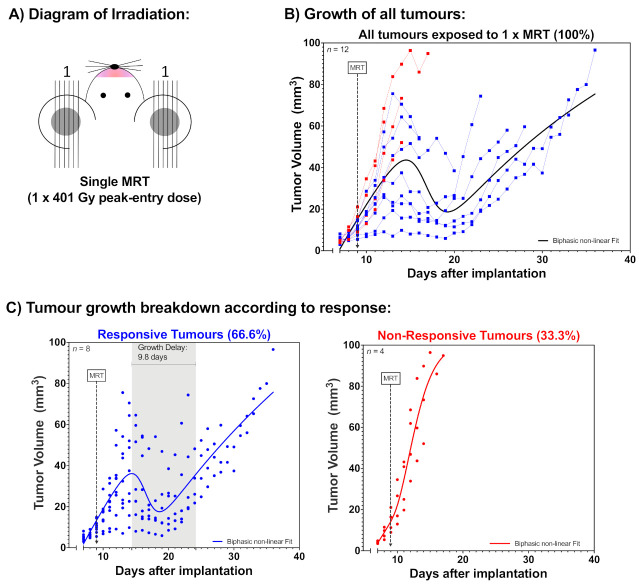
Tumour response after single Microbeam Radiation Therapy (MRT) (1 × 401.23 Gy). Melanomas in the ear of mice were exposed to a single MRT array of 401.23 Gy peak dose and 7.4 Gy valley dose, delivered nine days after tumour cell implantation. (**A**) Diagram of the irradiation modality; (**B**) Tumour growth of all melanomas after the single MRT; (**C**) Breakdown of the melanomas according to their treatment response; tumours exhibiting shrinkage and consecutive regrowth with a measurable growth delay were called “Responsive Tumours”, while melanomas that did not shrink after MRT were called “Non-Responsive Tumours”. A non-linear biphasic fit was used to model the growth.

**Figure 2 cancers-12-02656-f002:**
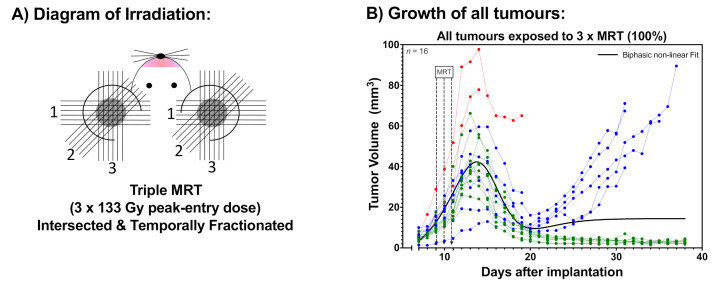
Tumour response after temporally fractionated MRT (3 × 133.41 Gy). B16-F10 melanomas were exposed to the three fractions of MRT with 133.41 Gy peak dose (2.46 Gy valley dose) per fraction. (**A**) Diagram of the irradiation modality, with the first (1), second (2), and third (3) MRT array intersecting at a 45-degree angle over 3 days; (**B**) Growth of all tumours after fractionated MRT. Irradiations occurred on days 9–11 after B16-F10 cell implantation; (**C**) Breakdown of tumours according to their response. Melanomas that underwent complete remission were called “Ablated Tumours”. Melanomas exhibiting tumour shrinkage and consecutive regrowth with a measurable growth delay were called “Responsive Tumours”, while melanomas that did not shrink were called “Non-Responsive Tumours”. A non-linear biphasic fit was used to model the growth.

**Figure 3 cancers-12-02656-f003:**
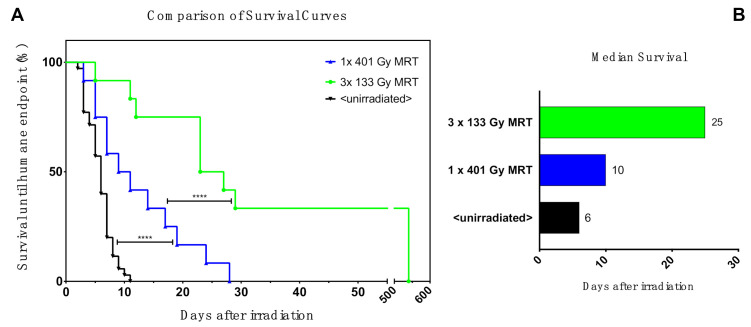
Analysis of animal survival after treatments. (**A**) Comparison of survival between the single (1 × 401.23 Gy peak dose), the triple fraction (3 × 133.41 Gy peak-dose) of MRT, and re-used data of unirradiated melanomas (*n* = 35) from our previous published experiment (<unirradiated>) [3]. Survival was defined as the time elapsed between the irradiation until the humane endpoints were reached. The differences were significant for the Gehan–Breslow–Wilcoxon test, which gives extra weight to the early timepoint: Chi-Square X^2^ (df = 2, *n* = 59) = 24.35, and *p* < 0.001. (**B**) Median survival. The median survival for the fractionated group is 23 days regardless of whether the mice showing tumour ablation are included in the calculation. Significant values of four orders of magnitude are represented with ****.

**Figure 4 cancers-12-02656-f004:**
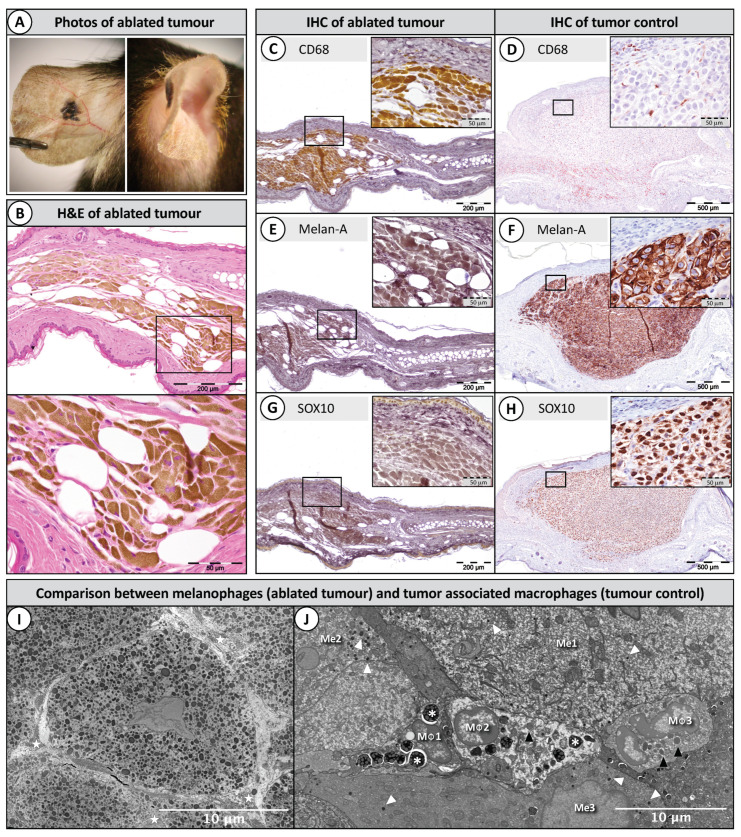
Histopathology of ablated tumours (3 × 133.41 Gy). (**A**) A picture of the remaining nevus 18 months after MRT. (**B**) The nevus was formed by melanin-laden cells; (**C**,**E**,**G**) the ablated tumours stained immunohistochemically (IHC) and immediately bleached with trichloroisocyanuric acid (TCCA) to eliminate the excess of melanin; positive staining is shown with the 3,3′-Diaminobenzidine chromophore (DAB) as a bright yellow colour. Melanin-laden cells were positive for the macrophage marker CD68 (**C**), while they were negative for two melanoma markers: Melan-A (**E**) and SOX10 (**G**); (**D**,**F**,**H**) are positive controls in a melanoma harvested 10 days after implantation (not bleached); positive staining is shown with the NovaRED chromophore; (**I**) the ablated tumour under transmission electron microscopy revealed that the melanophages were packed with melanin and separated by dense collagen fibres (stars); (**J**) a group of macrophages (MΦ 1, 2, 3) in between melanoma cells (Me 1, 2, 3) 6 days after a single MRT. Macrophages contain numerous primary lysosomes (black arrowheads) and secondary lysosomes (asterisks). Melanin granules are indicated with white arrowheads.

**Figure 5 cancers-12-02656-f005:**
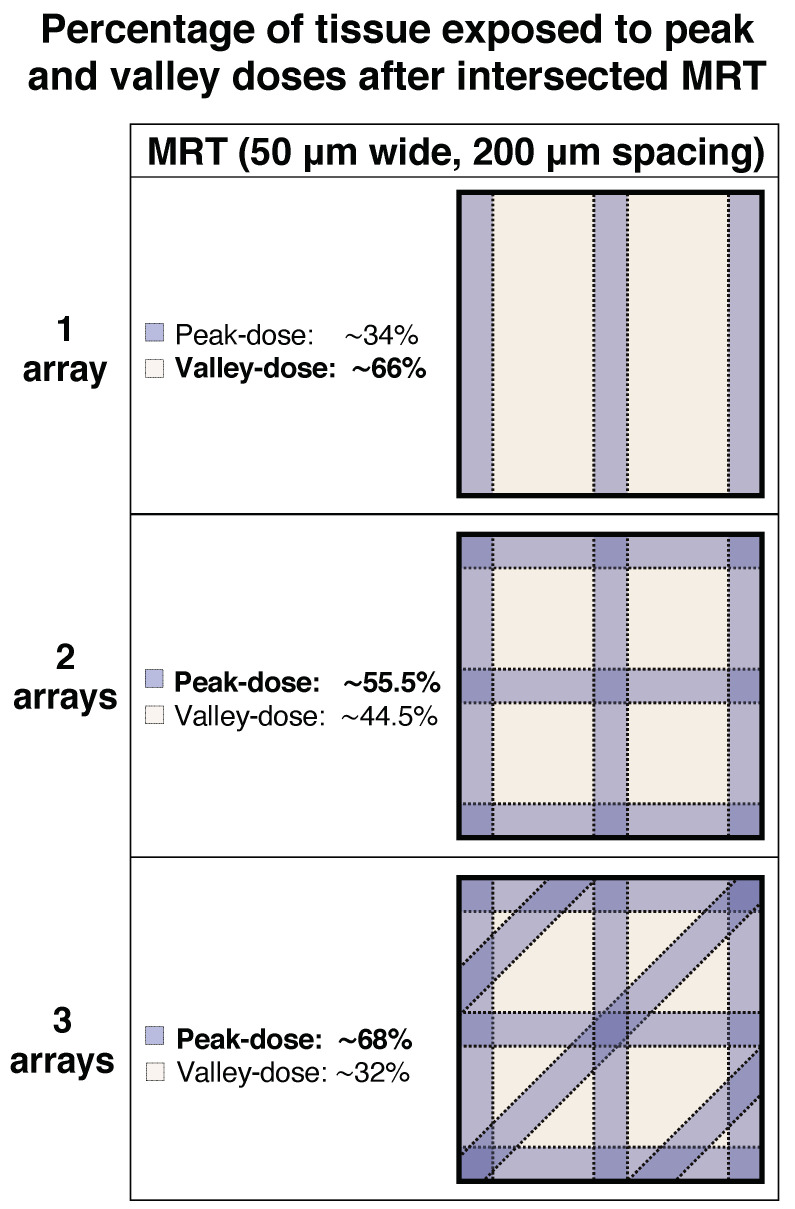
Intersecting the MRT arrays changes the total percentage of tissue exposed to peak and valley doses. This figure is a two-dimensional schematic representation of the geometrical distribution of three intersecting MRT arrays (microbeams 50 μm in width, 200 μm spacing) in a squared piece of tissue. As the number of intersecting arrays increases from one to three, the percentage of tissue exposed to peak doses increases in an inversely proportional matter to the percentage of tissue that remains exposed to valley doses.

**Table 1 cancers-12-02656-t001:** Experimental parameters.

Technical Parameters	Single MRT	Cross-Fired Temporally Fractionated MRT
Experimental Parameters:		
Sessions	1	3 (1 per day)
Peak dose in the tumour centre *	401.23 Gy	133.41 Gy
Valley dose in the tumour centre *	7.40 Gy	2.46 Gy
Peak to valley dose ratio (PVDR)	54.24	54.24
Number of microbeams	38	38
Microbeam width	50 µm	50 µm
Microbeam spacing (centre-to-centre)	200 µm	200 µm
Field size	7.6 wide × 8 high (mm^2^)	7.6 wide × 8 high (mm^2^)
Mean dose rate	11,704 Gy/s	12,870 Gy/s
Average irradiation time per slice	34.2 ms	11.4 ms
ESRF Parameters:		
Filling pattern	7/8 + 1	7/8 + 1
Current	171–168 mA	200–171 mA
Dose rate	68.99 Gy/s/mA	68.99 Gy/s/mA
Mean energy of spectrum	104 keV	104 keV
Peak energy of spectrum	88 keV	88 keV

* Monte Carlo simulation.

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
