# Peer review of "Complete Remission of Mouse Melanoma after Temporally Fractionated Microbeam Radiotherapy"

_cancers, 2020, doi:10.3390/cancers12092656_

Round 1

Reviewer 1 Report

The authors should check Fig 3A and correct a type there

'Com parison ofStructural Curves' to 'Comparison of Structural Curves'. 

Reviewer 2 Report

This paper investigated biological effects in microbeam radiation therapy (MRT) using a multislit collimator which provides 20-100 um width in the beam delivery. Moreover, this system achieves ultra-high dose rate beam delivery. Indeed, there is growing interest in biology in MRT compared with conventional radiation therapy. In this manuscript, the authors demonstrated usefulness of 3 fractionated microbeam radiotherapy (MRT) compared with single fraction dose. This is the first preclinical evidence and would be of interest for readers. However, there are some important deficiency especially on biological assessment and data interpretation which have to be improved.

Major points

(1) One of the key findings in this manuscript is that stronger antitumor response was obtained in 3 fraction form different angles in 3 consecutive days. However, the source of the differences in tumor response between single fraction and 3 fractions from different angles remains unclear in the current experimental scheme. It might be just because the 3 fractionated regimen reduces the dose reduction in the valley. Or it might be because fractionation in 3 consecutive days provided reoxigeneration as seen in conventional radiation therapy. The data of beam delivery from 3 different angles in one day will give us the answer. This experiment has to be performed to answer this fundamental question.

(2) What happens if more different angles of beams (eg; 6 directions, 9 directions, or more) are delivered? Is it possible to deliver microbeam without slit while keeping ultra-high dose rate? If it is possible, I guess, much stronger antitumor effect would be obtained because of the less collimated space. Please discuss these points.

(3) Fig 3.

Generally, statistical method has to be determined before experiments on the basis of hypothesis. Therefore either log-rank or Wilcoxson test should be selected.

(4)Fig 4

The authors evaluated histopathology only for ablated tumors. The same evaluation for responded and nonresponded tumors in single fraction group should be included to discuss about the differences in biological effects between single and 3 fractionated delivery.

(5) In the discussion, authors mentioned 3 fractionated regimens induced stronger immune response. However, they only evaluated macrophage. Again, since no comparison in macrophages was performed between single delivery and 3 fractionated delivery groups, this point cannot be discussed without direct comparison.

(6) The authors put a reference by Dewan et al. that demonstrated hypofraction but not single fraction of high dose induced stronger abscopal effect. This does not link the data of the authors because the finding of Dewan et al. is T cell mediated but not macrophage mediated phenomena shown in this manuscript.

Reviewer 3 Report

This is a very interesting and provocative report regarding the optimization of microbeam radiotherapy for eventual clinical benefit.    It is a very sound study comparing a single microbeam exposure to three fractions of microbeam, while keeping the total dose of each scenario constant.  The fractionated microbeam exposures obtained a clearly improved therapeutic outcome in terms of tumor control.  It is very exciting to have a demonstration of this increased efficacy and also produces new questions about mechanisms by which radiation may be working in this format.  There are a couple of areas that the authors have not considered in their discussion that would improve to scope and understanding of the work as well as several minor adjustments that may help the presentation of the data.  Overall, this is an excellent report and should be a valuable contribution to our growing understanding of FLASH and spatially fractionated radiation approaches.  

Specific comments:

1.  A major area that is not discussed is the relative benefit of the fractionated approach for tumors vs. the possible increase in normal tissue exposure and normal tissue damage.  Please add some thoughts on this for clinical feasibility in the discussion or introduction.

2.  A second variable/concept is missing from the discussion of possible mechanisms by which the fractionated treatment is attaining better tumor control.  There are a number of studies in the literature showing that microbeam or SBRT/GRID treatment changes the tumor physiology significantly with hours to days of the first exposure.  Please add some text regarding the possibility of increasing tumor oxygenation or reducing interstitial pressure being part of the increased response to the second and third fractions.   this would give the reader a bettter perspective and not solely  base the possible mechanism on the initial results the authors show with macrophage infiltration.  For ex.  2019 Radiation Research  PMID: 31188068 or similar articles could be cited.

3.  line 38:  suggest that the first sentence of the Introduction be re-worded to be more conventionally stated:  "....(MRT) is a pre-clinical approach of spatial fractionation that has been found to obtain superior tumour control...."

4.  line 42 and other places:  It is more conventional to state the distance between the beams in the array as 'center to center' distance.  Thus, please state as "...space 50-500 um from beam center to beam center".  This should be carried through to all places that you describe the distance between beams.

5.  line 58:   "...three fractions of 133.41 Gy"   It is stated this is a lower dose which might increase translation potential, but isn't this still an ablative radiation dose?  what are the lower limits of ablative dose, if known?

6.  line 68  "Two groups of 12 mice..."

7.  Discussion and throughout:   The conventional way to state survival time and things like radiation response is usually not is percentage, rather by using a normalized 'fold-change' schedule.  For ease of comprehension, please change the percentage reporting to fold-change (i.e., 250% = 2.5-fold)

8.  line 188:  "In another study...."

9.  line 201-202:  suggest restating as  "The implications of valley dose are and important variable to keep in mind since most valley doses would be considered significant in a radiobiology context on their own and future work focus on...."   10.  line 218:  'benefits'  Please state what they may be (i.e., reoxygenatioin, immune cell infiltration, normal tissue repair etc.)   11.  line 227-228:  The current study gave radiation over 3 days, and thus radioresistance would not have time to develop.  Therefore this statement is not relevant, or should be re-stated.      

Round 2

Reviewer 2 Report

Authors seriously responded most of my comments and concerns.However, some points have to be further specified.

(1) Authors responded to my question in the 1st bellow round but none of the points was not discussed in the manuscript. Please add the points in the discussion section or somewhere.  

(My first round review question) What happens if more different angles of beams (eg; 6 directions, 9 directions, or more) are delivered? Is it possible to deliver microbeam without slit while keeping ultra-high dose rate? If it is possible, I guess, much stronger antitumor effect would be obtained because of the less collimated space. Please discuss these points.

(2) The authors responded that histopathological analysis was not able to be performed for non-ablated tumor mice because of laceration of the skin. However, the analysis can be performed earlier time points before laceration of the skin. It has to be done in the next experiments and be specified as a future required study.
